# Research on Membranes and Their Associated Processes at the Université Paris-Est Créteil: Progress Report, Perspectives, and National and International Collaborations

**DOI:** 10.3390/membranes13020252

**Published:** 2023-02-20

**Authors:** Lassaad Baklouti, Christian Larchet, Abdelwaheb Hamdi, Naceur Hamdi, Leila Baraket, Lasâad Dammak

**Affiliations:** 1Department of Chemistry, College of Sciences and Arts at Ar Rass, Qassim University, Ar Rass 51921, Saudi Arabia; 2ICMPE, CNRS, Université Paris-Est Créteil, UMR 7182, 2 Rue Henri Dunant, 94320 Thiais, France; 3Department of Pharmaceutical Chemistry, Faculty of Clinical Pharmacy, Al Baha University, Al Baha P.O. Box 1988, Saudi Arabia

**Keywords:** membranes, UPEC university, membranes for energy, bleach, fouling and antifouling, Qassim university, KubSU university

## Abstract

Research on membranes and their associated processes was initiated in 1970 at the University of Paris XII/IUT de Créteil, which became in 2010 the University Paris-Est Créteil (UPEC). This research initially focused on the development and applications of pervaporation membranes, then concerned the metrology of ion-exchange membranes, then expanded to dialysis processes using these membranes, and recently opened to composite membranes and their applications in production or purification processes. Both experimental and fundamental aspects have been developed in parallel. This evolution has been reinforced by an opening to the French and European industries, and to the international scene, especially to the Krasnodar Membrane Institute (Kuban State University—Russia) and to the Department of Chemistry, (Qassim University—Saudi Arabia). Here, we first presented the history of this research activity, then developed the main research axes carried out at UPEC over the 2012–2022 period; then, we gave the main results obtained, and finally, showed the cross contribution of the developed collaborations. We avoided a chronological presentation of these activities and grouped them by theme: composite membranes and ion-exchange membranes. For composite membranes, we have detailed three applications: highly selective lithium-ion extraction, bleach production, and water and industrial effluent treatments. For ion-exchange membranes, we focused on their characterization methods, their use in Neutralization Dialysis for brackish water demineralization, and their fouling and antifouling processes. It appears that the research activities on membranes within UPEC are very dynamic and fruitful, and benefit from scientific exchanges with our Russian partners, which contributed to the development of strong membrane activity on water treatment within Qassim University. Finally, four main perspectives of this research activity were given: the design of autonomous and energy self-sufficient processes, refinement of characterization by Electrochemical Scanning Microscopy, functional membrane separators, and green membrane preparation and use.

## 1. Introduction

Membranes are mainly thin films that allow the permeation of various substances at different rates, and thus have the capacity to act as filters for the different components of solutions. The study of membranes as separators dates to the 18th century. At this time, their use was limited to laboratory work, and they had no place in industry because they were expensive, insufficiently selective regarding the substances they let pass, lacked durability, etc. [1]. The development of modern membrane science can be traced back to the 1960s at the University of California, when a reverse osmosis membrane was developed that was selective and reliable, whilst also being thin and able to maintain a high rate of solvent flow [1]. Since then, membrane technology has flourished, with custom-designed membranes for a diverse range of industrial, medical and water purification applications.

Two main families of membranes can be identified: porous membranes, which only allow molecules with a size smaller than the pore diameter to diffuse through, and dense membranes, where the transport occurs via absorption into and diffusion through the membrane [2]. Also, we can distinguish the charged membranes comprised—in addition to the matrix—of functional sites with partial or total charges from the neutral membranes which do not comprise of or have very few of these functional sites. 

Very often, dense membranes are more selective (prevent the passage of a larger number of molecules) but have the disadvantage of a significantly reduced flow rate, and vice versa for porous membranes. To optimize the operation of some processes, many membranes are engineered to include both of these properties, comprising a thin dense skin and a thicker porous layer, thus achieving a higher selectivity at the same time as maintaining sufficient flux [2,3,4].

These membranes, whatever their nature, separate two compartments: one called feed, which contains the solution to be treated, and the other permeate, to which the species move. To ensure the operation of these processes, a force is required to effectively push the species from the feed solution through the membrane. In the case of dialysis, the force driving the process is a difference in solute concentration between the two compartments. Electrodialysis is driven by the application of an electrical potential difference, while reverse osmosis, nano, and ultra-filtration are driven by a pressure gradient [3,4].

Membranes can be engineered from a range of materials, both natural (such as animal skins or organs, rubber, wool…) and synthetic (hydrocarbon, fluorocarbon, ceramic, composite…), depending on the intended function. In industry, synthetic polymer materials are the most frequently used, with the most common types coming from the cellulose acetate, polystyrene, polyfluorinated, polyester, and polyamide families of polymers [2], although advances in ceramic membrane design and production mean their use is becoming more widespread [5].

The range of membrane processes fulfilling industrial and medical applications is diverse. Reverse osmosis and nanofiltration [5] are used in the dyeing industry [6,7,8] and for the removal of alcohol from beer and wine [9,10,11,12]; dialysis processes [13,14,15,16,17,18,19,20,21,22,23,24,25,26,27,28], which have been used in medicine ever since the first functional dialysis machine was built in 1938 by Willem Kolff to restore electrolyte balance in blood and in water treatments; electrolysis processes for green hydrogen or Chlore/Alkali and bleach production; and electrodialysis, which is used in drinking water production and in the food industry [29,30,31].

One of the particularities of these processes is their ability to easily adapt to local problems and to the specificities of the solutions to be treated. However, their main limitation is the phenomenon of membrane fouling [32,33,34,35,36,37,38] induced by the accumulation of rejected material on or in the membrane. This has a significant negative effect on both the membrane selectivity and the flux through the membrane. So, these applications necessitate an energy increase and a chemical cleaning of the membrane to maintain the original flux rate. The latter requires more labor and cleaning agents, which may in turn reduce the membrane’s lifetime [39,40]. These two points (labor and cleaning agents) require more expenses, making membrane processes less economically competitive.

In parallel with the development of membranes and the processes using them, a metrological activity [9,41,42,43,44] for the characterization, diagnosis and autopsy of membranes was developed to accompany this development and try to understand its fundamentals [15,45,46,47,48,49,50,51,52,53,54,55,56]. Also, the scientific activity of modeling material structures [57] and/or matter transport phenomena through membranes [58,59] has flourished in the last four decades.

In this review we will present the main research and development activities on polymeric membranes and their associated processes at the Institute of Chemistry and Materials Paris-Est (ICMPE) of Thiais, a Joint Research Unit (UMR 7182) between the National Center for Scientific Research (CNRS) and the University Paris-Est Créteil (UPEC). After a brief history of membrane research at UPEC, we will focus on the major results obtained during this last decade in the new themes developed, the main old collaborations (Kuban State University) and recent ones (Qassim University), and the main insights we see for our research. This review will not only make it possible to summarize our work, but also to target other regional or international collaborations to break some technological locks still present.

## 2. Brief History of Polymeric Membrane Research at UPEC

Research on polymer membranes at UPEC started in 1970, when the university opened. The latter was called “Université Paris 12—Val de Marne”. The research team at the time was led by Professor Michel Guillou [60], who was also the creator and director of the University Institutes of Technology of Créteil (University of Paris 12) and Dakar (University Cheikh-Anta-Diop). A strong collaboration with Senegalese colleagues was initiated under his direction until his departure in 1975 for other ministerial and then high-level administrative responsibilities. The research activity was focused on the fractionation of liquid or gaseous mixtures by the pervaporation process. His successor, Professor Bernard Auclair, took over the management of the team, which became the Ion-Exchange Materials Laboratory (Laboratoire des Matéraiux Echangeurs d’Ions: LMEI), until September 2007. Recognized by the Ministry of Higher Education and Research as a “Welcome Team” (Equipe d’Accueil: EA 3120), the LMEI moved away from pervaporation processes to get closer to new promising themes at the time, namely the study of resins and ion-exchange membrane materials. This activity continued until 1 January 2009, when the LMEI became part of the Institut de Chimie et des Matériaux Paris-Est (ICMPE) of UPEC university. This integration was led by Professors Christian Larchet and Lasâad Dammak, with the objective of increasing the national and international visibility of research on polymeric membranes within UPEC. Indeed, the ICMPE has put many highly developed platforms in the service of this activity, such as microscopy, chromatography, spectroscopy, and spectrophotometry.

Until the beginning of the 1990s, research activities were focused on membrane metrology, with the central objective of determining the properties and physicochemical characteristics of resins and membranes under standard operating conditions and not necessarily similar to those in real operations. Very close links have been established between the LMEI, the AFNOR (Association Française de la Normalisation, key contribution to the standard NF X45-200, 1995) and some industrial companies such as EDF (Electricité de France), Rhône-Poulenc. The research activities have been progressively and continuously extended to the development of new simple processes for water treatment (mainly dialysis processes), and the modelling of the phenomena of matter transport and the microstructure of the membranes. During this last decade, other topics have been developed, such as multi-scale electrochemical characterization, the study of the membranes’ durability, the synthesis and applications of new composite membranes for energy (selective extraction of lithium, production of green hydrogen (a confidential subject which will not be dealt with in this article)), the production of certain products of first sanitary needs (bleach), and the treatment of certain industrial effluents.

Figure 1 below details the main dates, research themes and institutional and industrial collaborations of the polymer membrane activities at UPEC. The dates are given by years and not by months to simplify the presentation.

## 3. Main Results Obtained during the Period 2012–2022

Over the period of 2021–2022, the research on membranes within ICMPE was focused on four main themes, three of which concerned composite membranes (synthesis and use for the highly selective extraction of lithium, synthesis and application for the production of bleach, preparation of low-cost composite membranes for the treatment of water and industrial effluents), and the last one concerned ion-exchange membranes (deepening of the studies on their fouling and antifouling). Below, we will detail the context, the approach and the main results obtained, as well as future perspectives. 

### 3.1. Composite Membranes: Synthesis, Characterization, and Applications

The study of composite membranes at UPEC University is a rather recent activity since it only started in 2012 with the preparation of low-cost and fouling-resistant clay/PTFE membranes for wastewater treatment. Since then, this activity has not stopped growing in parallel with the development of theoretical and applied research activities on ion-exchange membranes. During the years 2021 and 2022, the activities on these two types of membranes became equivalent in terms of investment, results and scientific productions, as well as in terms of international collaboration with the Qassim University and the Kuban State University.

In this part we will not make a general and exhaustive bibliography on these membranes, but rather give our main results and draw up our own perspectives on different axes under study.

#### 3.1.1. Highly Selective Lithium-Ion Extraction

Lithium procurement has become a worldwide policy due to the rapid development of Lithium-Ion Batteries (LIBs) [61,62,63]. Its selective recovery from different sources is gaining important scientific and industrial interest [64]. One of the means to produce this element is its extraction from different aqueous sources (salars, geothermal water, etc.), which is advantageous compared to mineral ones due to its availability in solution [63,65]. Several methods were proposed, such as precipitation [66,67], adsorption [68,69], solvent extraction [70,71] and membrane technologies [72,73,74]. However, the presence of other mono and divalent cations makes this extraction relatively complex.

The main goal of our activity on this axis consisted of the elaboration of novel composite membranes presenting high selectivity towards Li-ions. The novelty of these membranes was the combination of Lithium Ion Conductive Glass Ceramics (LICGC) particles, an anionic exchanger polymer binder (PECH-DABCO+ NH_2_-PES), and a nonionic surfactant (Brij76) [75]. Thus, several preliminary studies have been carried out to optimize the membrane composition and its preparation method. The final composition of the prepared membranes is presented in Table 1.

These membranes were prepared by blending technique. They were characterized and showed a homogenous morphology, with a good LICGC particle dispersion in the polymer matrix and good thermal stability at up to 200 °C, as shown in the Figure 1. They presented significantly high Li^+^ ionic conductivity compared to Na^+^ (see Table 1).

These membranes were first tested in Diffusion Dialysis (DD) and seemed to be very selective towards Li^+^ compared to Na^+^ and K^+^. The best Li^+^ flux (30.35 × 10^−9^ mol.cm^−2^.s^−1^) and selectivity (S(Li/K) = 363 S(Li/Na) = 278) were found using LCM5. This optimal membrane has lately been used to investigate the effects of several experimental parameters (pH, co-ion nature, feed solution composition, treatment duration) by DD [76]. The schematic illustration of ionic diffusion through the LCM5 is presented in Figure 2. DD results have shown that the Li^+^ recovery rate reaches 22.1% of its initial feed concentration after 27 days of treatment, which is very efficient compared to the evaporation process. This membrane was also tested with Cross Ionic Dialysis (CID). CID-obtained results reveal that optimal Li^+^ diffusion is obtained when a solution of HCl (0.1 M) is used as receiver solution. The effect of the [Na^+^]_F_/[Li^+^]_F_ ratio in the feeding solution on LCM5 performance with DD and CID was also studied. It was found that a high recovered Li^+^ concentration was reached, even at a high ratio. At [Na^+^]_F_/[Li^+^]_F_ = 40 and after 24 h of treatment, it successfully extracted 16.63% of the Li^+^ with S(Li^+^/Na^+^) = 5543 by DD, and 36.78% of the Li^+^ with S(Li^+^/Na^+^) = 931 by CID.

The application of this LCM5 on electrodialysis was tested using two ED cells: a two-compartment and a four-compartment cell [77]. The second one was found to be advantageous thanks to its membrane-protection ability, presented in Figure 3. The effect of [Na^+^]_F_/[Li^+^]_F_ and the applied current density on membrane performances using a four-compartment cell was tested. The rise of the current density had a positive effect on the transfer of Li^+^ through the LCM5, while it had a negative effect on membrane selectivity towards Li-ions. By increasing [Na^+^]_F_/[Li^+^]_F_ ratios to 10, we improved the recovery ratio of Li^+^ and S(Li/Na). Nevertheless, for high values of this ratio, membrane performances were declined. Thus, the optimal results were obtained at a ratio close to where more than 10% of the initial Li^+^ concentration was extracted, with a selectivity value of around 112 after 4 h of ED experimentation at 0.5 mA.cm^−2^.

It should be noted that we have recently published a review which completes all this experimental work. In the referenced review [78], we collected up-to-date results on studies of traditional and promising methods for lithium recovery from natural solutions and extracts obtained from the disposal of spent batteries. The majority of the attention is paid to membrane methods. Known approaches are classified and analyzed, and the experimental and theoretical aspects of membrane-based ion separation are described; separation mechanisms and mathematical models are discussed. Baro- and electromembrane methods relatively that are well-developed at the laboratory level, which are used for the extraction of lithium from a mixed solution containing large amounts of Mg^2+^ and Ca^2+^ cations, are considered. The applications of commercial and lab-made membranes are analyzed. Novel and emerging approaches, which are promising for their effective separation of Li^+^ cation from a mixture of singly charged cations, are examined, including hybrid electrobaromembrane methods.

For perspectives, we aim to (i) enhance the separation performances by using a couple of other protective anion-exchange membranes with low proton leakage (for example PVDF-g-PDMAEMA based membranes), (ii) test the LCM5 for Lithium recovery from a real/natural source (for example the Salar de Atacama of Northern Chile) by DD, CID, and ED, (iii) study the adsorption of Li-ions by LCM, and (iv) study lithium recovery from spent lithium ion batteries (LIBs) by LCM5 membrane electrodialysis.

#### 3.1.2. Bleach Production

Membrane electrolysis, historically developed for the chlor-alkali process, is becoming more and more common as a replacement for mercury electrolysis devices. However, both the aggressiveness of some solutions to be treated and some of the products obtained (Cl_2_, ClO^−^, etc.) mean that the lifetime of the membranes is often quite short. It is therefore necessary to develop highly conductive, cheap, and chemically and thermally resistant membranes.

We have therefore developed a procedure that has allowed us to elaborate a new porous composite membrane based on polytetrafluoroethylene (PTFE) using an activated ceramic: boron nitride (BN). We have studied the effects of many experimental parameters: filler content, PTFE defibrillation time, temperature, and heating time, etc. After optimization of these operating conditions, we obtained a highly conductive membrane with good mechanical properties, excellent chemical resistance, and good thermal stability. This was subsequently confirmed using the following characterization techniques. 

SEM images of the surface and cross section of the BN/PTFE membrane in Figure 4 show that our composite membrane has a symmetrical morphology, with a thickness of 400 ± 10 µm. In this way, the ceramic particles are homogeneously distributed on the surface of the sample, and they give the membrane a slightly rough appearance. These particles, in most cases, are held together by a network of PTFE filaments, providing both good mechanical strength and a high percolation threshold.

The thermal properties of boron nitride and the BN/PTFE membrane were studied by thermogravimetric analysis (TGA) in an argon atmosphere. The following thermograms presented in Figure 5 allow us to conclude that our composite membrane exhibits high thermal stability since the greatest mass loss occurs only at high temperatures, compared to non-fluorinated polymer-based membranes.

Due to its chemical stability and resistance to aggressive agents, this composite membrane has been tested in the electrolysis of a saturated NaCl solution to produce, in one step, a commercial bleach (NaClO at 12 °Chl.) by using a three-compartment zero-gap electrolyzer (see Figure 6a). The separation between the two compartments is assured by pairs of membranes (ion exchange or composite) and by maintaining an acidic medium of which pH = 5, and we obtained a hypochlorous acid (HClO) solution on one side and a NaOH solution on the other side. A kinetic study on the performance of the hypochlorite generation process was carried out on different types of membranes by observing the results obtained in Figure 6b. However, after the neutralization of the HClO by concentrated NaOH, we obtained NaClO, but with a maximum content of 6 °Chl. The bleach thus obtained could be used in the disinfection of swimming pools and/or the degradation of organic pollutants. However, the limiting factor for the final NaClO content is the high porosity of the membrane.

The production of bleach at high concentrations by electrolyzing a saturated sodium chloride solution is not simple. Several parameters and factors are involved in both chloride oxidation and water reduction. Several secondary reactions can take place, disturbing the final chemical balance. Also, the aggressiveness of the hypochlorite ions and chlorine molecules is so important that few materials can resist them.

We have shown that it is possible to reach active chlorine contents close to 1.8% wt., which is significantly higher than the contents obtained by simple electrolysis with or without a separator (0.3 to 1.0% wt.), and that the limiting factor is the back diffusion of the HClO produced in the anode compartment into the other compartments through the pores of the ceramic membranes. For this reason, we are currently examining different approaches to reduce this porosity, either by using a finer ceramic powder or by adding nanoparticles (multilayered carbon nanotubes, graphene and its derivatives, etc.). These inert nanoparticles will reduce the diameters of the pores and improve the mechanical resistance of the membranes.

#### 3.1.3. Water and Industrial Effluent Treatments

Better production and less pollution are the challenges faced by industrialists in all sectors. With industrial development and population growth, water pollution is the result of wastewater discharge without any prior treatment or due to inefficient and non-environmentally friendly processes. The textile industry, mining, surface treatment, etc., discharge large quantities of chemicals harmful to health (carcinogenic dyes, heavy metals, oils, etc.) and cause pollution of surface water and groundwater. The treatment of the effluents of these industries has become an absolute necessity to preserve the environment and also to have a secondary source of water that rarifies from one year to the next due to climate change. This wastewater reuse treatment can take different forms: physical, thermal, chemical or biological processes, such as biodegradation, membrane filtration, chemical oxidation, ion exchange, electrochemical methods, adsorption and coagulation and flocculation. Below, we present some membrane treatments that we have developed at UPEC in collaboration with international universities.

(a)Development of ultrafiltration Kaolin membranes over Sand and Zeolite supports for the treatment of electroplating wastewater. Collaborations with Qassim University and University of Sfax.

Large volumes of oily wastewater discharged from industrial activities are considered one of the major contaminants to the environment and may pose a risk to human health [79,80]. In particular, oily wastewater generated from the electroplating industry is usually composed of an organic materials mixture and heavy metals, which are extreme pollutants worldwide [81,82]. These types of effluents exist in several countries such as France, Saudi Arabia, Tunisia, which makes this subject a point of common interest and collaboration between the membrane research teams of UPEC (France), the Qassim University (Saudi Arabia) and the University of Sfax (Tunisia).

Preliminary studies have led us to conclude that amongst different alternative technologies, membrane separation using porous ceramic membranes appears to be promising and efficient for the treatment of these pollutants. Natural kaolin has been utilized as the main raw material for preparing less costly membranes [83], thanks to its excellent strength, low plasticity, and good hydrophilicity to membranes [84].

The objective of this work was the development and the characterization of new composite UF membranes based on Kaolin over sand and zeolite supports. Natural Kaolin, collected from the Tabarka region (northwest Tunisia), was principally composed of a large quantity of silica SiO_2_ (55.25%) and alumina Al_2_O_3_ (24.17%). Moreover, the mineralogy analysis proved the presence of 61% kaolinite and 39% illite, while the non-clayey minerals were essentially represented by quartz [85]. The characteristics of the two tubular supports based on natural sand and zeolite, used for the preparation of the new composite membranes Kaolin/Sand and Kaolin/Zeolite [86,87], are summarized in Table 2.

Firstly, the sand and zeolite supports were rinsed with hot water and then with ethanol via ultrasound irradiation, to eliminate residual particles. After that, the cleaned supports were dried overnight at 100 °C. Then, two suspensions were prepared, differing by compositions previously optimized for the coating of sand and zeolite supports [87,88]:A total of 8% of Kaolin powder (ϕ < 53 µm) was mixed with 62% of water and 30% of PVA (12 wt% aqueous solution) for Kaolin/Sand membrane.A total of 2% of Kaolin powder (ϕ < 53 µm) was mixed with 68% of water and 30% of PVA (12 wt% aqueous solution) for Kaolin/Zeolite membrane.

Then the sand and zeolite supports were coated, respectively, via the Layer-by-Layer and Slip-casting methods described elsewhere [86,89]. Finally, the green membranes were kept in air for 24 h, followed by sintering in a programmable furnace before characterization and application. Thermal cycling was performed in two steps: the first, annealing at 250 °C for 3 h to eliminate the residual water and organic additives, and the next, annealing at 900 °C for Kaolin/Sand and 850 °C for Kaolin/Zeolite for 3 h to ensure the sintering of the membranes.

Referring to SEM results, it was found that UF kaolin layers are homogenous and exhibit good adhesion onto different supports (Figure 7a,b). In addition, the fabricated Kaolin/Sand and Kaolin/Zeolite membranes presented an average pore diameter in the range of 4–17 nm and 28 nm, as well as a water permeability of 491 L·h^−1^·m^−2^·bar^−1^ and 182 L·h^−1^·m^−2^·bar^−1^, respectively.

The oil and heavy-metal removal efficiency of these membranes were evaluated using a home-made plant under a temperature of 60 °C to reduce the viscosity of the effluent and a transmembrane pressure (TMP) of 1 and 3 bar for Kaolin/Sand and Kaolin/Zeolite, respectively. During the treatment of the industrial electroplating wastewater, there was a high stabilized permeate flux in a range of 306–336 L·h^−1^·m^−2^·bar^−1^, almost total oil retention, and a chemical oxygen demand (COD) removal of up to 96% was achieved. In addition, the fabricated membranes displayed a heavy metal removal efficiency of 99% for lead, 96% for copper and 94% for zinc (Figure 8). 

The relative goodness of the prepared UF kaolin membranes for the oily wastewater treatment was assessed by comparing its performance with ceramic Titania UF membrane (150 kDa) previously tested by Aloulou et al. [90]. All these ceramic membranes displayed similar oil rejection higher than 99%, and a high removal of heavy metals of up to 95%. However, the permeate flux of the commercial membrane of about 232 L·h^−1^·m^−2^ at an optimal transmembrane pressure of 3.5 bar was lower than that observed with kaolin membranes at lower applied pressures. The overall data suggest that the developed kaolin membranes have the potential for the remediation of oily industrial effluents contaminated by heavy metals.

The main aims of the next study are the preparation, characterization and application of mixed matrix membranes, i.e., polymer/inorganic composite membranes, using low-cost and natural materials for the treatment of industrial wastewater.

(b)Polymer inclusion membrane processes for industrial effluent treatment. Collaborations with the University of Gafsa.

In the past decade, polymer inclusion membrane processes have become an interesting alternative to conventional solvent extraction methods for the selective separation of compounds such as acids and metals from industrial and waste effluents [91,92]. These membranes are easy to make and have good mechanical characteristics. 

This work reports on the preparation of a novel polymer inclusion membrane (PIM) composed of CTA, di(2-ethylhexyl) phosphoric acid and lignin, added as matrix modifier.

The lignin used in this work was acetylated with acetic anhydride in the presence of pyridine as a catalyst, leading to acetylated Kraft lignin (AKL), in accordance with the procedure reported by Cachet et al. [93]. CTA and AKL (15% wt.) were dissolved in CH_2_Cl_2_ under continuous magnetic stirring to reach a homogeneous solution. After vigorous stirring, the carrier D2EHPA was added, and the solution was stirred for one hour. This solution was then poured into a 8.0 cm diameter Petri dish, and the organic solvent was allowed to evaporate overnight [43]. The film was then carefully peeled out from the dish (Figure 9).

Prepared membranes were characterized by attenuated total reflectance Fourier transform infrared spectroscopy (ATR-IR). IR spectra illustrate that there is no interaction between the CTA/AKL membrane and D2EHPA [94,95]. The surface and cross-section morphology of the membrane was inspected by scanning electron microscopy (SEM) (Figure 10). The CTA/AKL/D2EHPA membrane presents a homogeneous dense structure with no apparent porosity. The pores of the CTA membrane had been filled by the D2EHPA molecules yielding to this dense structure. The cross-section image of the PIM 40 confirmed that D2EHPA acting as a plasticizer for the CTA/AKL matrix was uniformly distributed in the matrix of the CTA, creating a sort of liquid pathway [96,97,98]. 

Measurements of the contact angle and the tensile strength were realized to have information on hydrophobicity and mechanical properties of the PIM. Young’s modulus was increased with 40% wt. of D2EHPA loading. This result approves the plasticizing effect of D2EHPA already noted in the literature [99].

In this study, the prepared membrane was used for the recovery and separation of the metals that come from the rinsing baths of surface treatment industries. The effluent is charged with Cu(II), Ni(II) and Cr(III). The best membrane is enhanced when the concentration of D2EHPA is 40% wt. The influence of the feed phase pH and the strip phase composition on removal performance was explored. Results indicate that the percentage of extraction is highly dependent on the pH of the feed solution, and the pH value is 5 for maximum separation. The influence of the nitric acid concentration in the strip solution was examined, and it was found that the percentage of extraction and stripping is further increased due to the difference in the driving force. The results indicated that PIM was able to extract 74% of the Cu and 55% of the Cr over only 20% of the Ni (Table 3). Similar results were found for extracting Cr(III) with D2EHPA [100].

Going forward, we want to use other extractants for the extraction of other pollutants. The synthesized membranes can be tested in the field of gaseous effluent treatment.

(c)Removal of micro-pollutants by Donnan Dialysis. Collaborations with Qassim University and University of Tunis.

The pollution of nitrogen (N) and Phosphorus (P) is due to the growth of the population, industrialization, and rapid urbanization. This pollution is often in the form of nitrate, nitrite and phosphate, and presents a water quality problem [101]. The eutrophication of water supplies and infectious diseases are the main environmental problems due to the presence of nitrate and phosphate. This phenomenon is responsible for the dramatic growth of algae occurring in internal and coastal waters [102]. Consequently, nitrate, nitrite and phosphate removal from wastewater can be an effective method for controlling eutrophication. Nitrite is more dangerous than nitrate because it is very toxic to human health. The World Health Organization (WHO) considers a nitrate concentration of 50 mg/L in irrigation water and a nitrite concentration of 0.5 mg/L in drinking water to be the minimum levels [103] to anticipate harmful effects of high concentrations of nitrates and nitrites on human health.

Chemical process and physicochemical processes such as coagulation–flocculation were used to remove nitrate, nitrite and phosphate; they are not efficient for removing them both. The choice of the use of Donnan dialysis was essentially economical. Donnan dialysis is a continuous low-cost process, as it requires only a few chemicals, pumping energy, and it is easy to handle. The cell of the Donnan dialysis device was in a Plexiglas. It comprises four parts assembled by means of threaded rods. The centering of the different parts is ensured by shoulders. The end caps have a bore in which a star-shaped magnetic stirrer is housed. The bar is driven by a rotating magnetic field, generated outside the cell. The two central compartments are bored over their entire length by a hole, and their centering is ensured by a shoulder; the membrane introduced into the bottom of the shoulder seals between these two elements. Receiver and feed tanks with a 1000 mL Erlenmeyer flask were used to supply the two cell compartments using a controlled peristaltic pump by a variable speed drive, and they were fitted with two identical pumps, ensuring equal flow rates in the two compartments. Fluid circulation takes place in flexible pipes. Donnan dialysis is a membrane process that consists of cross-ion exchanges having the same electric charge through an ion exchange membrane between two solutions. The driving force in Donnan dialysis is the chemical potential gradient. Donnan dialysis is widely used to recover and eliminate various ions such as nitrates, fluorides, chromium and phosphates [14,21,104,105,106,107,108,109].

To investigate the simultaneous removal of nitrates, nitrites and phosphate by Donnan dialysis, a statistical tool was applied. The Response Surface Methodology is an efficient statistical strategy to design experiments, build models, determine optimum conditions, and evaluate the significance of factors and even the interaction between them. The Doehlert design is considered the efficient one, and the matrix has the advantage of being flexible; the possibility of increasing experimental domains or adding new factors to the design without having to repeat the experiments that have already been demonstrated. 

Three anionic exchange membranes have been used: Neosepta^®^ ACS, Neosepta^®^ AMX and Neosepta^®^ AFN. These membranes are homogeneous, and they have the same structure properties as polystyrene/divinylbenzene. The AFN and ACS membranes were generously provided by Eurodia Industries S.A., and the AMX one was purchased. In Table 4, the characteristics and properties of the anionic exchange membranes are presented.

Before the use of the anionic exchange membranes, a treatment was used according to the French standard NF X 45-200 (1995) [110]. The treatment consisted of the preparation and conditioning of the membranes.

First, the removal of one component in the feed compartment was performed with different parameters, such as the choice of counter-ion, concentration of counter-ion in the receiver compartment, and concentration of nitrate, nitrite, and phosphate separately in the feed compartment to determine the experimental field (Figure 11). 

We focused on the choice of the counter-ion; the main constraints were the high mobility, the non-toxicity for the environment, the low cost of the electrolytes, and the non-reactivity with the other components of the solution to be treated. According to the obtained results, NaCl was the best counter-ion.

After the choice of the counter-ion, it was important to determine the concentration of counter-ions. The effects of the chloride concentration in the receiver compartment—varying from 0.001 to 0.5 mol.L^−1^—on the removal rates of the nitrate (Figure 12a), nitrite (Figure 12b) and phosphate (Figure 12c), were evaluated for the three tested membranes (AFN, AMX and ACS).

This is explained by the fact that increasing the concentration gradient between the feed solution and the counter-ion compartment and the transfer of phosphate, nitrate, and nitrite (separately) from the feed compartment to the receiver compartment is easier, and a better percentage of removal is obtained.

Afterward, the removal of nitrate, nitrite, and phosphate in the feed compartment by the Response Surface Methodology (RSM) using the Doehlert design was investigated to optimize the process and to understand the simultaneous transport of nitrite, nitrite, and phosphate. All the experiments were performed by commercial membranes such as AFN, ACS and AMX.

In order to improve the removal of nitrate, nitrite and phosphate, the synthesis of the membranes will be conducted from biopolymer to develop and design blended pectin/alginate hydrogel membranes. The choice of the matrix is purely economic, since it is extracted from brown seaweed that is available on the French and Tunisian coasts.

We will start with the preparation of the alginate/pectin membranes, which is carried out first by the gelation of the alginate by the addition of EPTAC, which is prepared using a conventional solution casting technique.

DRX, FTIR, SEM analyses of the membranes will be carried out. Then, a preliminary study of Donnan dialysis and the start-up with synthesized membranes and their application to phosphates, nitrites, and nitrates separately and simultaneously will be done after setting the appropriate operating conditions according to the RSM model.

Finally, a study of the aging of synthesized membranes will be applied to better understand the causes and consequences of the aging of MEIs following use in a membrane process, which necessarily requires a comparative and expanded study of the state and behavior of the new and used samples. At this stage of the study, all information that can be collected is pertinent to achieve our objective.

### 3.2. Ion-Exchange Membranes: Characterization, Applications in Dialysis Processes, Fouling and Antifouling Studies

#### 3.2.1. Ion-Exchange Characterization

Membrane characterization is one of the main steps in the development process of new membranes or applications. This characterization can have different aspects: mechanical, physicochemical, electrochemical, thermal, or structural. Most often, it is done ex situ under standardized conditions defined by standards. In recent years, methods of in situ and operando characterization have appeared. The objectives of this characterization, if possibly global, are multiple:To evaluate the individual performance of a new membrane.To be able to compare membranes between them.To evaluate the effects of a desired modification (surface or mass treatments) or not desired (fouling, scaling) on the performance of a membrane,To understand the often-complex relationships between the microstructure and overall performance.

Historically, the LMEI of UPEC was one of the major actors for the implementation of the French Standard NF X45-200 [110], which is still used by many international laboratories and by industrialists. During the last decade, UPEC has developed two interesting techniques for the characterization of ion exchange membranes: the membrane ionic conductivity clip, which is an ex situ technique, and the Scanning Ion Conductance Microscopy (SICM), which is an in situ and operando technique. We will briefly describe these two techniques.

(a)Clip cell for membrane conductivity

The development of membrane processes using an electrical potential gradient (electrodialysis and its derivatives) encourages the correct definition of the electrical conductivity of this type of functional polymer. The methods of evaluation of this conductivity have not ceased to diversify—so much so that the results obtained can be contradictory. We have measured this dynamic characteristic with three different methods for three ion exchange membranes of distinct natures [51]. The three measurement methods are the LMEI clip cell [46,111], the mercury cell and the Guillou’s cell [60]; as for the membranes, two were based on sulfonated and cross-linked polystyrene (CM1 and CM2), and one was based on sulfonated and unbridged polytetrafluoroethylene (Nafion^®^117). The measurements were carried out in sodium chloride medium for concentrations ranging from 0.1 to 2 mol L^−1^. The results show a dependence on the chosen method. We have remarked that the order of the curves representing the conductivity changes from one method to another, which justifies the need to know the limits of each of them. The LMEI has adopted a method with two electrodes, avoiding any contact with the membrane. It seems, from the completed studies, that it is the most adequate for the domain of moderate and high concentrations of the equilibrating electrolyte. We have found that, for the high concentrations, the conductivity values corresponding to the three methods are almost equal, but for the low concentrations, the differences can be significant. The results given by the Guillou’s cell are always inferior to those obtained by the two other methods. This is due to a concentration polarization and to the fast formation of two diffusion boundary layers at the membrane–solution interfaces. These layers constitute additional resistances to the ionic transfer. As for the mercury cell, although recognized and currently used by Russian searchers, it has shown little differences with the LMEI clip. We have attributed these small disparities to the impedance at the membrane–mercury interfaces or to the contamination of the membrane by the mercury.

It was following this comparative study that we undertook, within our team at the UPEC, the improvements carried out for the improvement of the circulation of the solutions and gases in the two compartments of this clip (see Figure 13). Thus, we created openings in these compartments so that the gases formed—especially for low concentrations where the measurement requires more time—and can escape easily, inducing a local agitation and a renewal of the solution in each of the compartments. These modifications have significantly improved the quality of the measurements in terms of accuracy and repeatability. These modifications were the subject of the French patent n° FR1601694, entitled “Electrical conductivity clip”.

The LMEI clip has many advantages: no use of mercury; not expensive; portable and movable at the production site; a wide range of frequencies; real conditions of use for the media; non-destructive control of the media; accessibility to different points of the media; and a wide range of concentrations and electrolytes, etc.

(b)Scanning Ion Conductance Microscopy. Collaboration with Kuban State University

It is known that ion-exchange membranes have inhomogeneities (consist of phases with clear interfaces). In the case of so-called homogeneous membranes consisting of an ion-exchanger and a reinforcing fabric, it does not exceed hundreds of nanometers [112]. In this case, heterogeneous membranes (or composite ones), consisting of at least an ion-exchanger, an inert binder and a reinforcing fabric, have a micrometer scale. At the same time, it was found in a number of works that screening a part of the membrane surface can lead to an improvement in its properties due to the enhancement of over-limiting convective ion transport mechanisms [113,114,115]. The reinforcing fabric is made from materials that are non-conductive to ions, which also leads to the appearance of inhomogeneities. In addition, using optical microscopy, it was found that the reinforcing fabric affects the shape of the membrane surface and can give them waviness, which also affects the ion transfer rate [116].

Classical methods for studying the properties of ion-exchange membranes have a number of shortcomings. The main difficulty is that for an objective assessment of the inhomogeneity parameters, the membrane must be in a swollen state, since the size of a weakly cross-linked ion-exchanger can increase several times during swelling. Another difficulty is the close light conductivity of the ion-exchanger and the polyethylene, which also greatly affects the accurate determination of the size of the conductive and non-conductive areas. At the same time, the ion-exchange material protruding on the surface can be shielded by a thin film of an inert binding agent and does not participate in the ion transfer process, and an almost imperceptible crack can provide access for ions to the ion-exchanger inside the membrane, which also makes it difficult to objectively assess the conductive properties of the IEM. The study of waviness and roughness (or, in the general case, geometric heterogeneity) is hampered by the limitations of the methods; atomic force microscopy allows for research at the nanometer scale, and profilometry often provides information on roughness only along one direction.

All of the above difficulties in visualizing the IOM surface morphology can be solved by using more suitable research methods, for example, scanning probe microscopy. Within the framework of Franco–Russian cooperation, a device for the characterization of composite ion-exchange membranes that implements the Scanning ion conductance microscopy (SICM) method was developed and tested (Figure 14).

The method makes it possible to visualize ion currents flowing through the channels of porous materials and heterogeneous materials containing ion-conducting sections and duplicating the surface morphology of the material under study within a certain distance. The scanning probe is made in the form of a microcapillary filled with an electrolyte solution, the tip of which is usually located at the distance of its radius from the surface under study. By controlling the change in the ion current depending on the position of the probe, a feedback signal is created [117,118].

In the case of ion-exchange membranes, scanning is carried out directly inside the electrodialyzer during its operation [119]. An inverted U-shaped frame is installed in the desalination chamber, equipped with an outlet to prevent the overflow of the solution (Figure 15). Through this frame, the mobile capillary has access to the surface of the studied membrane fixed between the usual frames of the electrodialyzer.

The distribution of the electric potential near the surface of the swollen CMX obtained by the SECM technique is shown in Figure 16. When scanning, the normal distance from the mobile microcapillary to the undulated conductive surface changes. As a result, the potential drop between the fixed and the mobile measuring capillaries changes as well [120]. Since the conductivity of the 0.02 M NaCl solution is lower than that of the CMX membrane [121,122], the potential drop is smaller near the hills and higher near the valleys. This allows you to determine the parameter *b* (Figure 16), which is the distance between the top of the hill and the bottom of the depression on the surface, which correlates with the classical roughness parameters (Ra, Rz, etc.).

The distribution of the electric potential drop near the surface of the commercial heterogeneous cation exchange membrane MK-40 is uneven (Figure 17) [123]. The characteristic sizes of the conductive sections on the surface of the commercial membranes range from 5 to 50 μm [124] and differ in shape, which leads to an extremely uneven distribution of the potential drop. Nonetheless, the centers of the conducting areas or their agglomerates are uniquely determined from the peaks of the potential drop. It is clearly seen that there are extensive regions on the membrane surface, with the largest potential drop associated with the non-conductive regions.

#### 3.2.2. Neutralization Dialysis for Brackish Water Demineralization

The neutralization dialysis (ND) process was first suggested for water desalination by Igawa et al. [125] It is based on the simultaneous use of two Donnan dialysis processes at the cation exchange (CEM) and at anion exchange (AEM) membranes that separate the desalination compartment from the acidic and alkali chambers, respectively. The salt ions are removed from the desalination compartment during ion exchange through the membrane, and the H^+^ and OH^−^ ions enter this compartment as their substitutions. The H^+^ and OH^−^ ions recombine in the desalination compartment with the formation of water molecules. This chemical reaction is the main driving force of the method.

Literature data analysis shows that the application of ND to the desalination of the electrolyte solutions was studied mostly experimentally [23,24,125,126,127,128,129,130,131]. In early works [125,126,127,128], the ND process was realized with flow-through [125,126,131] and non-flow [128] cells of various configurations (flat sheet and spiral wound), but the experimental conditions were far from those used in electrodialysis (low solution flow rate [125], small volumes of treated solution [128]), which makes the comparison of the two processes a delicate problem. The effect of the different physicochemical parameters of the ND system on the rate and degree of the desalination of the strong binary electrolyte solutions was investigated. It was found that the tenfold increase in concentrations of the acid and base leads to a threefold increase in the flow of salt ions from the desalination compartment of the ND apparatus [128].

An attempt to optimize the ND process was made by conducting a series of experiments with three varied factors: initial concentration, flow rate of the acidic and alkaline solution, and the volume of the treated solution [24]. Statistical analysis of these data showed that the initial concentration of the base and the volume of the treated solution have the greatest influence on the duration of the ND process for strong electrolytes.

The ND method was also successfully used for the separation of weak acids and bases [126]. It was established that the rate of transport of such substances increases with an increase in their dissociation constant. It was also shown that the transport rate of amino acids (glycine) depends on the pH in the desalination compartment, and is at maximum at the isoelectric point (for glycine pH = 6) [130,131]. 

Hybrid systems for the desalination of a strong electrolyte solution are also being developed, combining neutralization dialysis and capacitive deionization [129]. It has been shown experimentally that the efficiency of the hybrid device is 20% higher than that of a separate ND device, but compared to ND, its disadvantage is the need to apply an external electric field.

As for the theoretical description of the ND process, in the pioneering work by Denisov et al. [132], the ion transport was modeled on a system consisting of three compartments (acidic, alkaline and saline) and two ideally selective homogeneous ion-exchange membranes. The resulting one-dimensional quasi-steady state model does not take into account the membrane structure and the hydrodynamic conditions in the compartments of the system. The model parameters were determined by fitting. This model made it possible to reveal the presence of pH fluctuations in the desalination compartment, depending on the initial acid and base concentrations. 

Relatively simple mathematical equations were suggested, allowing for the estimating of the flux through the membrane depending on the concentration of acid (base) and salt in steady state [23,132]: (1)$$J^{c}=\frac{D_{H}^{c}D_{Na}^{c}X^{c}}{d^{c}\left(D_{H}^{c}-D_{Na}^{c}\right)}\ln\frac{\left(D_{H}^{c}K_{H,Na}c_{H}^{s,A}+D_{Na}^{c}c_{Na}^{s,A}\right)\left(K_{H,Na}c_{H}^{s,D}+c_{Na}^{s,D}\right)}{\left(D_{H}^{c}K_{H,Na}c_{H}^{s,D}+D_{Na}^{c}c_{Na}^{s,D}\right)\left(K_{H,Na}c_{H}^{s,A}+c_{Na}^{s,A}\right)}$$
(2)$$J^{a}=\frac{D_{OH}^{a}D_{Cl}^{a}X^{a}}{d^{a}\left(D_{OH}^{a}-D_{Cl}^{a}\right)}\ln\frac{\left(D_{OH}^{a}K_{OH,Cl}c_{OH}^{s,B}+D_{Cl}^{a}c_{Cl}^{s,B}\right)\left(K_{OH,Cl}c_{OH}^{s,D}+c_{Cl}^{s,D}\right)}{\left(D_{OH}^{a}K_{OH,Cl}c_{OH}^{s,D}+D_{OH}^{a}c_{OH}^{s,D}\right)\left(K_{OH,Cl}c_{OH}^{s,B}+c_{Cl}^{s,B}\right)}$$
where $D_{i}^{c}$ and $D_{j}^{a}$ are the ion diffusion coefficients in CEM (*i* = H^+^, Na^+^) and AEM (*j* = OH^−^, Cl^−^), respectively; *X* is the membrane exchange capacity; *d* is the membrane thickness; *K* is the ion exchange equilibrium coefficient; *c* is the ion concentration in solution; upper index *s* is the membrane surface; and *A*, *B* and *D* means belonging to acid, base and desalination compartment side.

It was found [23] that the moment of transition of kinetic control from internal diffusion (the transport is limited by the membrane) to external diffusion (transport is limited by the diffusion boundary layers) shifts in time, depending on initial concentration of acid and base, and also on solution flow rate. In [127], it was shown that the performance of the desalination increases with a decrease in the width of the desalination compartment.

Using a non-steady state model [15] of the ND process in batch mode, it was found that the oscillations in the pH saline solution are linked to the delay in the variations of the concentration profiles in the membranes after changing saline solution composition. When ion exchange across one of the membranes is dominant, this generates strong pH variations in the solution, which accelerates the ion exchange through the other membrane. However, the passage of the “leadership” from one membrane to the other requires some time, during which the pH continues to vary. The increasing rate of ion exchange across the other membrane results in a slight change in the direction of the pH variation.

The application of this model revealed effects which cannot be described using a steady state approach. Such effects are caused by the initial state of CEM and AEM (as a result of the membrane’s history, i.e., the membrane preparation procedure or the previous ND process conditions) before the beginning of the ND process. It was found that in the case where the CEM is in the H^+^ form and the AEM is in the OH^–^ form in the beginning of the ND process, high exchange rates across the surface layers of the membranes bathing the saline solution result in a few pH fluctuations of the saline solution (Figure 18). 

These fluctuations cannot be described by the quasi-steady state model. However, the two models effectively describe the pH fluctuations occurring in the saline solution after the concentration profiles in the membrane approach a linear form.

Taking into account the dependence of the diffusion coefficients in the desalination solution on the ionic strength of the solution allows for the explanation of the occurrence of a small maximum in the kinetic dependence of the saline solution electric conductivity (Figure 19).

ND requires the use of an alkali solution, with which the AEM is in contact. The continuation of the model [15] became a non-steady state model, where the loss of exchange capacity of an AEM as a result of the deprotonation of the weakly basic functional groups was taken into account [26]. 

It was shown theoretically and experimentally that the fraction of strongly basic functional groups in AEM has a significant effect on pH and the rate of solution desalination. This result is important in the practical application of the ND. The mechanisms that lead to a decrease in the rate of desalination and to the acidification of the saline solution when the anion-exchange membrane loses a part of the exchange capacity (Figure 20) are revealed. 

Thus, the deprotonation of weakly basic functional groups in an AEM leads to an increase in the resistance of ion diffusion from the solution to the membrane and vice versa. A decrease in the effective exchange capacity causes a decrease in the Donnan exclusion of co-ions. As a result, the concentration of co-ions in the AEM increases significantly, causing an increase in the flux of these ions into the desalination compartment, and the desalination process is slowed down. 

For the case of an NaCl solution in the desalination compartment of an ND setup, the gradient of the Na^+^ ion concentration between the desalination compartment and the acid compartment increases, causing an increase in the exchange rate across the CEM. The Na^+^ ions in the desalination compartment are replaced with much more mobile H^+^ ions, and as a result, at the beginning of the ND process, an increase in the electric conductivity of the saline solution and a sharp decrease in its pH are observed (Figure 20). Due to the significantly higher exchange rate through the CEM compared to through AEM, the concentration of H^+^ ions remains excessive, and the gradient of their concentration between the desalination compartment and the acid compartment decreases, causing a decrease in the exchange rate across the CEM. At the same time, the flux through the AEM remains low and changes relatively slightly during the entire considered time of the ND process. When the salt concentration in the desalination compartment decreases, the ion transfer begins to be limited by external diffusion kinetics. Due to the fact that the mutual diffusion coefficient of the OH^−^ and Cl^−^ ions in the solution is about 10% greater than the mutual diffusion coefficient of the H^+^ and Na^+^ ions [23], the ion flux through the AEM+diffusion boundary layer (DBL) becomes higher than that through the CEM+DBL, causing a decrease in the concentration of H^+^ ions.

Thus, from the very beginning of the ND process (several tens of seconds) with an anion-exchange membrane containing a significant fraction of weakly basic functional groups, the deprotonation of these groups causes an increase in the diffusion resistance of the membrane. Due to the reduced effective exchange capacity, the Donnan exclusion of the co-ions decreases. As a result, the co-ion concentration in the AEM essentially increases, causing an increase in the flux of these ions towards the desalination compartment hindering the desalination process (Figure 21).

However, numerical experiments show that the proportions between weakly basic functional groups in AEM have a negligible effect on the above-mentioned saline solution characteristics due to a relatively high rate of deprotonation as compared to the duration of the ND process.

#### 3.2.3. Fouling and Antifouling of Ion-Exchange Membranes

Electrodialysis (ED) was first established for water desalination and is still highly recommended in this field for its high water recovery, long lifetime, and acceptable electricity consumption. Today, thanks to the technological progress of ED processes and the emerging of new ion-exchange membranes (IEMs), ED has been extended to many other applications in the food industry. This expansion of uses has also generated several problems, such as IEMs’ lifetime limitation due to different aging phenomena (because of organic and/or mineral compounds). If these aspects are not sufficiently controlled and mastered, the use and efficiency of ED processes will be limited, since they will no longer be competitive or profitable compared to other separation methods. The current commercial IEMs show excellent performance in the ED processes; however, organic foulants such as proteins, surfactants, polyphenols or other natural organic matters can adhere to their surface (especially when using anion-exchange membranes (AEMs)), forming a colloid layer or infiltrating the membrane matrix, which leads to the increase in electrical resistance, resulting in higher energy consumption, lower water recovery, loss of membrane permselectivity and current efficiency, as well as lifetime limitation.

In 2010, our team launched a new research project on the fouling of IEMs used in different applications in the ED process. This research axis was imposed following feedback from an industrial partner who did not understand the premature ageing of certain IEMs. We therefore started with diagnostic phases of different IEMs extracted from ED stacks at different periods of its operation, which allowed us to start making hypotheses, which were confirmed or eliminated by laboratory tests, with an accelerated aging process. We then reproduced, on laboratory cells of the PCCell type [133], not only the electrochemical operating conditions of the MEIs, but also the cleaning operations of the stack. Then, we formalized all this by making modifications on the microheterogeneous model (work done in collaboration with Russian colleagues). Finally, we were able to propose customized solutions to reduce the fouling phenomenon of MEIs. These solutions were adapted to the nature of the treated effluents and to the operating conditions of the stack. Thus, we now have a mature and global experience, but still in perpetual evolution, on the techniques of diagnosis, accelerated aging, modeling, and antifouling. In addition to the numerous articles published on this topic, in 2021 we published two complementary journals on this topic. In the first one, we reviewed a significant amount of recent scientific publications, research and reviews studying the phenomena of IEM fouling during the ED process in the food industry, with a special focus on the last decade. We first classified the different types of fouling according to the most-used classifications. Then, the fouling effects, characterization methods and techniques, as well as the different fouling mechanisms and interactions and their influence on IEM matrix and fixed groups were presented, analyzed, discussed and illustrated [33]. One of the new and significant results in this review was the classification given by Table 5, which was established following the critical study of all the techniques of analysis and diagnosis of fouling.

In the second review, we analyzed modern scientific publications related to the effect of foulants (mainly typical for the dairy, wine, and fruit juice industries) on the structural, transport, mass transfer, and electrochemical characteristics of cation-exchange and anion-exchange membranes. The relationship between the nature of the foulant and the structure, physicochemical and transport properties, and behavior of ion-exchange membranes in an electric field was analyzed using experimental data (ion-exchange capacity, water content, conductivity, diffusion permeability, limiting current density, water splitting, electroconvection, etc.) and modern mathematical models. The implications of traditional chemical cleaning were considered in this analysis, and modern non-destructive membrane cleaning methods were discussed. Finally, challenges for the near future were identified [134].

Most of the “classical” [135,136,137,138] and modern [139,140,141,142,143] models are intended to describe transport phenomena in ideally homogeneous IEMs. The microheterogeneous model developed by Gnusin, Nikonenko, Zabolotskii et al. [59,144] seems to be the most suitable for real membrane systems. It is widely used in the literature [145,146,147,148,149,150,151,152,153,154] to interpret the experimental concentration dependences of membrane conductivity and diffusion permeability, and to determine the relationship of the “structure–transport properties” of IEMs. The model is based on a simplified representation of the IEM structure, according to which the membrane can be considered as a multiphase (in the simplest case, two-phase) system. The properties of such a system are determined by the properties of the individual phases, which are consistent with the fundamentals of the effective medium theory [155]. The gel phase (denoted by the index g) is a microporous swollen medium. It includes a polymer matrix carrying fixed groups, as well as a charged solution of mobile counterions (and, to a lesser extent, co-ions), which balance the charge of the fixed groups. There are two regions within the gel phase. A “pure” gel consists of an electrically charged double electrical layer and side polymer chains with hydrated fixed groups. The hydrophobic regions are formed by agglomerates of polymer chains that do not contain fixed groups, as well as an inert binder and reinforcing cloth. In the second phase, the inter-gel space (denoted by index “int”) is an electrically neutral solution (identical to the external solution). It is located in the central part of the meso- and macropores, and also fills in the membrane structural defects. A membrane fragment that includes both phases is shown in Figure 22.

In accordance with the approach that combines the effective medium theory and nonequilibrium thermodynamics [156], the flux of ions of sort *i* (*j_i_*) in a two-phase IEM is proportional to the gradient of the electrochemical potential $d\mu_{i}/dx$ [59,157]:(3)$$j_{i}=-L_{i}^{*}d\mu_{i}/dx$$
where $L_{i}^{*}$ is the effective conductivity coefficient characterizing a multiphase medium, which is equal to
(4)$$L_{i}^{*}={\left[f_{1}{\left(L_{i}^{g}\right)}^{\alpha }+f_{2}{\left(L_{i}^{int}\right)}^{\alpha }\right]}^{1/\alpha }$$

The conductivity coefficients $L_{i}^{g}$ and $L_{i}^{int}$ refer to the gel phase and the inter-gel solution phase, respectively. The structural parameter *α* takes values from 1 to −1, respectively, with a parallel and sequential arrangement of phases; the sum of the volume fraction of the gel phase (*f*_1_) and the inter-gel space (*f*_2_) is equal to one (*f*_1_ + *f*_2_ = 1). The values $L_{i}^{g}$ and $L_{i}^{int}$ are determined by the ion diffusion permeability coefficients $D_{i}^{g}$ and $D_{i}^{int}$, and ion concentration *C* in the corresponding phase:(5)$$L_{i}^{g}=D_{i}^{g}C_{i}^{g}/RT$$
(6)$$L_{i}^{int}=D_{i}^{s}C_{i}^{s}/RT$$

The exchange capacity of the membrane gel phase (the concentration of fixed groups in the gel phase), *Q_g_*, is related to the membrane exchange capacity, *Q*, by the equation: *Q_g_ = Q/f*_1_. From Equations (3)–(6), simple expressions are obtained to determine the membrane conductivity (*κ_m_*), diffusion permeability (*P_m_*) and counterion transport numbers (*t_m−_*) (for example, in AEM):(7)$$\kappa_{m}=\left(z_{+}L_{m+}+z_{-}L_{m-}\right)F^{2}$$
(8)$$t_{m-}=\frac{L_{m-}}{L_{m-}+L_{m+}}$$
(9)$$P_{m}=2t_{m-}L_{m+}\frac{RT}{C}$$
where index “+” relates to a cation (co-ion in the AEM).

The simple relationship between the membrane conductivity (*κ_m_*) and the conductivities of the gel phase (*κ_g_*) and electroneutral solution (*κ_s_*) filling the inter-gel spaces is of the greatest interest:(10)κm=f1κgα+f2κintα1α

If |*α|* is not too far from zero (<0.2), and the external solution concentration is in the range of 0.1 *C*_iso_ < *C* < 10*C*_iso_ (*C*_iso_ is the electrolyte concentration at the isoconductance point: *κ_m_* = *κ_g_* = *κ_s_*), Equation (10) may be approximated as
(11)$$log\left(\kappa_{m}\right)\approx f_{2}\log\left(C\right)+const$$
where const ≈ (1 − *f*_2_) log(*κ_g_*).

It is believed that in all the cases under consideration, fouling leads to a decrease in the membrane IEC. In the case of the formation of hydraulically permeable reticular colloidal structures (milk whey) [158] in the pores,
(12)$$L_{i}^{int}=\left(\gamma D_{i}^{s}\right)C_{i}^{s}/RT$$
where *γ* is a coefficient reflecting the ratio of ion mobility in the intergel space of a fouled membrane and in an external solution.

In the case of contact of IEMs with polyphenol-containing solutions (wine, juices, tea extracts), it is believed that dense hydraulically impermeable conglomerates of colloidal particles (denoted by index “*cp*”) are formed in the meso- and macropores, but do not penetrate the nanopores. These changes in the structure can lead to a possible change in the ion transport path, which is manifested in a change in the structural parameter from α to β. In this case, from Equation (10), it is easy to obtain an equation for *κ_m_* [41]:(13)$$\kappa_{m}={\left[f_{1}\kappa_{g}^{\alpha }+f_{2}{\left(f_{2s}^{int}\right)}^{\alpha /\beta }\kappa_{s}^{\alpha }\right]}^{1/\alpha }$$

Equation (13) can also be approximated in the same way as Equation (10):(14)$$log\left(\kappa_{m}\right)\approx f_{2}{\left(f_{2s}^{int}\right)}^{\alpha /\beta }log\left(C\right)+const$$

A similar approach was used to describe the transport characteristics of IEMs with conglomerates of various nanoparticles immobilized in their meso- and macropores [58].

The use of the basic [59] and modified microheterogeneous model [41,58,158] develops a theoretical basis for explaining the effect of foulants on the membrane transport characteristics and makes it possible to predict the tendency of IEM to fouling, depending on membrane structure and component composition of processed solutions in the food industry.

An IEM’s conductivity, as a rule, decreases after prolonged exposure to liquid media typical for the food industry [9,41,159,160,161,162,163]. In the case where a membrane is in contact with amino acids, carboxylic acids or anions of polybasic inorganic acids, the decrease in conductivity is not dramatic [160,162] and can be reversible if a dense layer of proteins or mineral precipitate does not form on the IEM surface [164,165,166]. The decrease in *κ_m_* is most often caused by electrostatic interactions and the hydrogen bonds of foulants with fixed EIM groups.

The *f*_2_ values of IEMs increase because of the polymer matrix stretching when strongly hydrated ions enter it [167], or due to its destruction as a result of cleaning (see Section 2), as well as due to the operation of membranes in intensive current regimes. For example, Garcia-Vasquez et al. showed that after several thousand hours of the Neosepta AEM (Astom, Osaka, Japan) used in the ED demineralization of whey, *κ_m_* decreases by a factor of 1.3, and *f*_2_ increases by 25%, as compared to pristine AEM.

In the case of polyphenol-containing liquid media, the membrane conductivity (and other transport characteristics) undergoes more significant changes. For example, after exposure to green tea, the *κ_m_* of the AEMs dropped by at least 50% (AMX-Sb, AFN, Astom, Osaka, Japan, and PC-400 D) [168,169]. After contact with wine and cranberry juice, the *κ_m_* of the AEMs decreased by a factor of 1.6–1.9 [162], 4 [9] (AEMs Neosepta), 3 [161] (MA-41P, Shchekinoazot, Pervomaisky, Russia) and 4 (AMX, AMX-Sb, Astom, Osaka, Japan). The conductivity of CEMs decreased by 2 [9,163], 1.1 (MK-40, Shchekinoazot, Pervomaisky, Russia), 1.2 (Fuji CEM Type II, Fujifilm, The Netherland; CSE-fg, Astom, Osaka, Japan) and 1.7 (CJMC-5, ChemJoi, Hefei, China) [159]. The decrease in conductivity is primarily caused by the loss of IEM in exchange capacity due to not only electrostatic, but also to π-π (stacking) interactions of foulants with ion-exchange materials.

It is important to note that the experimental data treatment on conductivity using a microheterogeneous model demonstrates a clear decrease in the volume fraction of the gel phase *f*_2_ in the case of CEMs. This result provided additional evidence for the formation of agglomerates of the hydraulically impermeable colloidal particles in the pores of cation exchange membranes.

As for AEMs, in some studies [161,162], an increase in *f*_2_ is observed, while in [9], the volume fraction of the intergel space decreases. To clarify the reasons for this apparent inconsistency, Bdiri et al. studied the fouling kinetics of aromatic AEM soaked in synthetic solution, which contained tartaric, acetic, lactic acids, KCl, CaCl_2_ and a high concentration of polyphenols (5 g.L^−1^) sufficiently greater than that in juices and wine [170,171]. Processing these data (Figure 23) using a modified microheterogeneous model, they showed that both the conductivity and the volume fraction of the gel phase decreased with an increase in the time of membrane contact with the phenol-containing solution (Table 6). 

Pristine membranes which have not been soaked in the model solution (h = 0) do not contain colloidal particles. However, *f*_2s_ = *f*_2_ = 0.09 values are slightly lower than the apparent volume fraction of the inter-gel spaces, *f*_2app_ = 0.11 (the slope of the log(*κ_m_*) vs. log(*C*) dependence, Figure 14). Colloidal particles are formed in the inter-gel spaces with an increase in soaking time. They displace a part of the electroneutral solution that results in decreasing *f*_2*s*_, and, therefore, *f*_2*ap*p_. A part of AEM functional groups becomes blocked by the colloidal particles that lead to IEC decreases with soaking duration. The increase in the counterion diffusion coefficient in gel phase $D_{-}^{g}$ (Table 1) should be due to an increase in membrane swelling degree. This parameter can be estimated by the ratio of the fouled and pristine AEM thickness, *d/d_h_ =* 0. Higher swelling indicates an increase in the size of the micropores that promote counterion mobility. Apparently, a higher swelling of IEMs in the considered liquid media is caused by relatively small and highly hydrated [172] organic acid species, such as tartaric acid.

Faster destruction of anion-exchange membranes during cleaning in ED industrial processes and in the case of application of intensive current modes is due to the more alkaline media inside AEM compared to CEM (see Section 2). If an increase in *f*_2_ caused by this factor is not compensated by a decrease in the inter-gel space due to the formation of agglomerates of colloidal particles, we observe an increase in the volume fraction of the gel phase and an increase in the AEM diffusion permeability. Note also that the diffusion permeability of AEMs in juices and wine may increase slightly if colloidal particles do not form agglomerates (no cleaning) and are completely or partially destroyed by concentrated saline solutions, which are usually used to study diffusion permeability [58,120]. The main types of interaction of polyphenols with ion-exchange membranes and the effect of this fouling on the transport characteristics of IEMs are summarized in Figure 24.

## 4. Main Collaborations

### 4.1. National Collaboration

The collaborations at the national level are of two types, collaborations with universities and industrial collaborations. One of the first collaborations set up by Professor Michel Guillou’s team was with the Laboratory of Filled, Reactive and Chiral Polymers and Biomimetics: ERA 471, which in 1983 became the Laboratory of Polymers, Biopolymers, Surfaces (PBS: UMR 6270 CNRS—University of Rouen), directed at the time by Professor Eric Seligny [173]. This collaboration was reinforced after the creation of the LMEI. Another very intense and dynamic collaboration was established between the LMEI and the Laboratory of Materials and Membrane Processes (UMR 9987 CNRS—University of Montpellier), which in 1994 became the European Institute of Membranes (IEM of Montpellier: UMR 5635 [174], directed at the time by Professor Claude Gavach and then by Professor Gérald Pourcelly. Following internal reorganization within the LMEI, as well as within the PBS of Rouen and the IEM of Montpellier, these two collaborations have gradually faded away in favor of other international collaborations (see below).

However, the UPEC membrane team has been able to build and maintain numerous collaborations with French industrialists, such as Eurodia Industrie S.A. [175], Neusca [176] and Gen-Hy [177]. The details of these industrial collaborations remain confidential, but we can say that they concern scientific cooperation to solve industrial problems related to the fooling of ion exchange membranes during electrodialysis processes and the production of very efficient and stable membranes to produce green hydrogen. These collaborations provide the team with important funding to maintain the exceptional level of equipment of our platforms.

### 4.2. European Collaboration

In July 2022, a new collaboration, in the form of a partnership, between the Belgian company AZA Battery and the membrane team of UPEC was born. This cooperation, which is very confidential, has been focused on the study of the possibilities of further improving the Zinc-Air battery technology, which is already very well developed by AZA Battery.

### 4.3. International Collaborations

#### 4.3.1. Collaboration with Kuban State University (KubSU) in Russia

The cooperation between our research team at the UPEC and the similar team at the KubSU includes several activities, among them:Implementation of joint research projects;Co-supervision of PhD students;Mobility of young researchers;Joint organization of and participation in scientific conferences;Joint participation in juries for the defense of dissertations.
(a)Implementation of joint research projects

The main joint scientific activity of the ICMPE is aimed at generating new knowledge in the material science of dense and nanoporous ion-exchange membranes and the physiochemistry of ion and solvent transfer in ion-exchange systems under operation. The research strongly involves mathematical modeling. 

The main scientific results concern the joint development of novel techniques for the characterization of ion-exchange membranes (IEMs) and the study of the mechanisms of intensive mass transfer in electromembrane systems.

French–Russian scientific cooperation started in 1995, when the first joint INTAS project (Nb 95-0026) was supported by EC. KubSU from Russia and Membrane Technology Enterprise from Kazakhstan were involved. The project coordinator was Prof. C. Larchet, LMEI—UPEC. Later, four other international INTAS projects and bilateral (one PICS and several PECO) projects were performed. On this basis an Associated International Laboratory (LIA—MEIPA), « Ion exchange membranes and associated processes » was created in 2010; its activity lasted up until 2018. The project envisaged not only joint research, but also the mobility of young scientists too. Note that in April 2013, at the meeting of the Supervisory Board of the National Research Center (CNRS) in Paris, the Russian–French LIA “Ion-exchange membranes and processes” was recognized as the best among 27 laboratories of this type in France. With that, a MemBridge Project (Nb 233253) FP7-NMP-2008-CSA-2 aimed at creating a global Membranes network was launched in 2009 and finished in 2011. FP7-Marie Curie project Nb 269135, titled CoTraPhen—Coupled Ion and Volume Transfer Phenomena in Heterogeneous Systems: Modeling, Experiment and Applications in Clean Energy, Micro-Analysis and Water Treatment. The start of the project was in 2011, its duration being 4 years. A relatively big PHC KOLMOGOROV PROJECT N° 38200SF lasted from 2017 to 2019. 

In Appendix A, a more detailed description of these projects is given in the end of this review.

The joint research activities resulted in the publication of more than 100 joint articles. Some of them contributed significantly to membrane science. Thus, the paper by Nikonenko, V.V., Pismenskaya, N.D., Belova, E.I., (…), Pourcelly, G., Larchet, C. 2010 Advances in Colloid and Interface Science, 160(1–2) 101–123 [178] was cited more than 263 times; the paper by Volodina, E., Pismenskaya, N., Nikonenko, V., Larchet, C., Pourcelly, G. 2005 [124] was cited more than 212 times; and the paper by Larchet, C., Nouri, S., Auclair, B., Dammak, L., Nikonenko, V. 2008 [157] was cited more than 107 times (according to Scopus, December 2022).

(b)Co-supervision of PhD students

The first program (Franco–Russian network program TRAINING—RESEARCH 1997–1999 (ref 1408/AP/LN/MJ, decision n 96P0079; case n 190518K grant n 190528G/bergere) was launched in 1997 under the direction of Prof. C. Larchet, LMEI—University Paris 12. During this program, three Russian students completed and defended their thesis in France (at the University of Paris 12, at the European Membrane Institute, Montpellier, and at the University of Rouen)—two students from Krasnodar and one student from Voronezh State University. 

Later, thanks to the support of the French Embassy, this activity was revived. Ms. Ekaterina Belashova (now Melnikova) did her thesis on IEMs under the supervision of Prof. G. Pourcelly and Dr. P. Sistat, and defended it in October 2014; the title was “Electroseparation of complex solutions for the production of organic acids: transport phenomena and reactions at the membrane/solution interfaces”. Ms. Marina Andreeva did her co-supervised thesis at ICMPE-KubSU, entitled “Experimental and theoretical study of mineral and organic clogging of ion exchange membranes during water treatment by electrodialysis or Donnan dialysis” from 2014. The joint supervision was performed by Professor L. Dammak (ICMPE) and Professor N. Kononenko (KubSU). During her study, Ms. Marina Andreeva published four articles under the French–Russian co-editorship.

Together with Russian students, an ICMPE student, Mrs Wendy Garcia-Vasquez, was supervised within the French–Russian LIA. She defended her thesis in 2014 and is working presently at the Eurodia Co., a French society and a world leader in the application of electro-membrane technologies.

(c)Mobility of young researchers

This kind of activity was carried out within all the projects, especially within LIA, MemBridge, CoTraPhen and PHC KOLMOGOROV. In addition, a great number of small individual projects supported by the French Embassy were carried out. It is rather difficult to consider all the exchanges; within CoTraPhen alone, there were about 50 person-month exchanges in both senses. The total number of person-month exchanges is about 100.

(d)Joint organization of and participation in scientific conferences

A total of 18 international conferences, French–Russian seminars and workshops were organized jointly. The most important event among them was the 44th International Conference on Ion Transport in Organic and Inorganic Membranes, held 21–26 May 2017 in Sochi, Russia. There were about 160 participants from nine different countries, including Prof. A. Yaroslavtsev (IGIC RAS, Moscow), the Chairman of the Scientific/organizing Committee, Prof. G. Pourcelly (IEM, Montpellier) was the Co-chairman of the Conference, Profs. M. Cretin (IEM, Montpellier), L. Dammak and D. Grande (UPEC), as well as Profs. V. Nikonenko, N. Kononenko, N. Pismenskaya (KubSU, Krasnodar), the members of the Scientific/organizing Committee.

There were four presentations from French colleagues. Prof. G. Pourcelly (IEM) made a plenary lecture, Dr. C. Trellu gave an oral presentation, and two PhD students, T. Ounissi and S. Bousbih, from UPEC, gave oral presentations within a special session in the ‘YOUNG RESEARCHERS CONTEST’. Separately, Russian colleagues gave their presentations on behalf of Russian and French laboratories. In total, four French and four joint French–Russian oral presentations and four posters were presented at this Conference. 

As for the conferences in France, a short conference was held at ICMPE, Thiais: Workshop Collaboration within French-Russian International Associated Laboratory “Ion-exchange membranes and related processes”, on 30 October 2018, with the participation of two Russian colleagues. This event was held the same day as the successful defense of the PhD thesis of Ms. Myriam Bdiri, who was a PhD student under joint supervision between the ICMPE (Thiais) and the Tunis University. As for the Russian participants, Victor Nikonenko was the president of the jury, and Dmitrii Butylski (PhD student) was in the research stay at the ICMPE, financed by the Mechnikov scholarship program.

(e)Joint participation in juries for the defense of dissertations

The defenses of all Russian PhD students were passed, with the participation of Russian colleagues in the jury. This included the defenses of Mr. S. Parshikov (1998) and Ms. M. Andreeva (2017) both at the UPEC. As well, the defense of the thesis of Ms. M. Bdiri at the UPEC (2018) took place with a strong participation of the LIA researchers in the jury, and especially Prof. V. Nikonenko. Also, the defense of the thesis of Mr. V. Nichka at KubSU (2022) took place, with the participation of Prof. L. Dammak.

It is worth noting that Ms. L. Jansezian, a student at UPEC, has obtained a three-month internship at MIT within the group of Prof. J. Han at MIT, USA, under the joint supervision of Prof. L. Dammak (UPEC) and Prof. V. Nikonenko (KubSU). At this time, a joint French–Russian–American article [179] was published.

#### 4.3.2. Collaboration with the University of El Qassim (Saudi Arabia)

The collaboration between UPEC and Qassim University is relatively young, since it only started in 2018 with a common project on the synthesis and cation complexation of p-tert-butyl-calix [4]arene bearing two 8-hydroxyquinoline units [180]. This collaboration has been strengthened in the last two years, as the two teams have combined their efforts on the theme of water treatment and on whether to make it drinkable or to depollute it.

In the water potabilization axis, a very ambitious project concerned demineralizing surface water in a way that is autonomous in terms of chemical products and energy, and this was by coupling 3 processes:The central process of Neutralization Dialysis (ND) [181] which is a process developed within the UPEC, and which allows for the substitution of the cations of a charged water with H^+^, and at the same time, the anions of this water by OH^−^ ions. The latter combines with the H^+^ to give water. Thus, the water initially charged with mineral salts is demineralized.The Bipolar Membrane Electrodialysis (BPED) process [182,183,184], which is used to produce caustic soda and hydrochloric acid by the electrolysis of cooking salt. These two acidic and basic solutions produced will be used to feed the ND process with H^+^ and OH^−^ ions. The BPED process must be powered by a low-voltage DC power source.The production of electricity by Photovoltaic Panels, which allows for the production of the direct current necessary to the operation of the BPED process and the pumps necessary for the circulation of the fluids. These panels will thus ensure the energy autonomy of the whole.

This project received a very favorable opinion from the two academic partners, but credit has not been allocated because of the earmarking of the credit lines for research against COVID. It is anticipated that this project will be reactivated in 2023.

In the axis of water depollution, Qassim and UPEC University have been working together for 6 years on this theme of common interest. Indeed, Saudi Arabia, and more particularly the central region of “Ar Rass”, has been struck for years by the scarcity of water and wishes to examine all the technically simple and financially affordable solutions to reuse water resulting from certain industries. Therefore, we have jointly developed studies to investigate the treatment of different wastewater [16,90,185], and especially water from the textile industry or from car washes, in order to make it reusable (e.g., for household tasks, irrigation, pre-washing, etc.). These applications are wide-spanning and interest almost all countries.

It should be noted that this activity has also brought together many colleagues from two Tunisian universities (University of Sfax, and University El Manar), who have expertise in the preparation of new composite membranes at a low cost and using some local products (clays). These membranes used in nanofiltration processes have shown good performance and have allowed for the reasonable cost of wastewater treatment processes in their regions.

This activity is still ongoing and is being developed more and more. Pilot implantations are planned for 2023 at Qassim University.

## 5. Main Perspectives

Our background, knowledge and know-how accumulated over more than half a century, and our current collaborations and the equipment available at ICMPE allow us to project ourselves into the next decade with great optimism, serenity, and motivation to work on the three topics briefly presented below. Other collaborations remain to be built that would go further to deepen these topics.

### 5.1. Design of Autonomous and Energy Self-Sufficient Processes

Many remote areas in many countries suffer from a lack of drinking water, even though they often have access to brackish water sources or sea water. This need for drinking water is also important following natural disasters or various regional conflicts. The solutions that currently exist (RO, NF, ED) are excellent solutions for very large-scale production, but they require a heavy investment, require a large power source and a highly skilled workforce, and cannot be installed quickly or moved from one location to another. In 2015, a Neutralization Dialysis process for surface water potabilization was developed at ICMPE. This process requires the use of acidic and basic solutions with concentrations close to 100 mM. To avoid the problems of cost, transport, and storage of these solutions, we propose to produce them on site using electrodialysis with the bipolar membrane of a solution of sea water or surface water. Electrical energy will be provided by a photovoltaic panel.

This project is the final step for building a mobile, self-sufficient unit for producing potable water for consumption in remote areas (which will be the subject of a national project). This is a great societal, scientific, technological, environmental, and financial challenge. It affects three issues at once: public health, primary public service, and land-use planning through population settlement. It aims to provide drinking water to a population that is often fragile and lacks the necessary financial and human resources. In this project, we couple a bipolar membrane electrodialysis process (EDMB) to electrical energy production by photovoltaic panels to produce acid and basic solutions, not too concentrated but sufficiently rich in H^+^ and OH^−^ ions to perform a Neutralization Dialysis operation. This coupling has never been done before.

The production of electricity for this application using photovoltaic panels has several advantages: the current produced is already continuous, ecological, easy to implement in different configurations, does not necessarily require strong sunlight and is easy to store. Even better, instead of storing electrical energy in batteries, we will take advantage of moments of sunlight to produce the acid and base solutions.

We aim to achieve the following objectives:

In the long term:To develop a mobile unit, autonomous in products and energy, to produce drinking water from sea water or brackish water. This unit will be sized for a large family (~10 people) or a village (~100 people).To ensure the efficiency of this unit.Short term:To produce acidic and basic solutions of molarity close to 100 mM from surface or sea water and from the electrical energy produced by photovoltaic panels.To study the durability and the robustness of the system.To estimate the production cost of these solutions and the carbon impact of this technique.

### 5.2. Refinement of Characterization by Electrochemical Scanning Microscopy

We have already developed, in close collaboration with our colleagues from Kuban State University, an innovative technique for the in situ characterization of ion exchange membranes during operation in an electrochemical process such as electrodialysis. This is the Electrochemical Scanning Microscopy technique, which has been patented [186].

This new technique has the advantage of not taking the membrane out of its operating environment, and of having a mapping of the membrane surface with a resolution in the order of 5 to 10 µm. A prototype is currently installed in the ICMPE-UPEC University laboratory. In June 2021, we resumed work on this prototype to make it more compact, easily movable, and with a better resolution with an objective of 1 to 2 µm.

### 5.3. Functional Membrane Separators

Membrane biofouling during electrodialysis processes is a severe issue for industries. It is caused by the attachment of microbial cells at their surface, followed by cell growth and multiplication generating a film which will impair the process efficiency by increasing energy consumption and reducing membrane lifetime. Ion-exchange membranes used in electrodialysis processes have a complex structure with a surface charge, hence the difficulty to provide antibacterial properties to membrane surfaces without impacting their charge. Thus, to avoid their alteration, we chose to modify membrane spacers, which are in direct contact with them. Membrane spacers are simple grids of polymer, and they were obtained by 3D printing using tailored-made filament extruded in our laboratory.

Membrane spacers were first obtained via process 1. Pellets of Polyamide 11 (PA11), a biosourced polymer extracted from castor oil, were mixed with magnesium oxides (MgO) directly in the filament extruder. The spool obtained was used to 3D print membrane spacers. A second process was developed by adding a pre-extrusion step. Pellets and powder were thus mixed in a twin screw extruder with a back flow channel. The pellets obtained were used in the filament extruder, which gave a spool used in the 3D printer. Processes are outlined in Figure 25. Spacers were characterized using SEM and EDX, and DSC, ATG, tensile test and antibacterial assays were performed.

With SEM imaging, we were able to visualize the surface of the membrane spacer and magnesium particles distribution, confirmed by EDX spectra. In Figure 26, we can see that raw PA11 spacers are smooth and do not contain magnesium. PA11 + MgO 1% wt. spacers, obtained via process 1, contain some magnesium particles, but they are rare, and the EDX signal is weak. Finally, PA11 + MgO 10% wt. spacers, obtained via process 2, contain a lot of magnesium particles with a strong EDX signal. Process 2 allows for a better distribution of particles and, more importantly, for reaching a higher content of magnesium. Indeed, the filament extruder cannot mix pellets with magnesium oxide powder content higher than 1% weight because of frictions with the screw. In process 2, using the first step of extrusion allows for the decrease in friction in the filament extruder. In addition, this first step also enables a more homogeneous mixing between magnesium oxide and polymer, leading to a better distribution along the filament.

Antibacterial assays were done on raw PA11 and PA11 + MgO 5% wt. (process 2) spacers. It appears (Figure 27) that adding magnesium oxide in membrane spacer allows for the reduction of the bacterial growth of *S. Aureus*, but no effect was observed on *E. coli*.

In-depth antibacterial assays will be conducted to find the optimum composition. In addition, MgO release will be studied by ICP, and biofilm formation will be quantified under static conditions with wastewater. Other treatments will be investigated, such as grafting biosourced polyphenols, which are known to possess antioxidant and antibacterial properties.

### 5.4. Green Membranes

The last area we have started to work on during these past two years (2021 and 2022) is the manufacturing of ion-exchange or porous-green membranes. This topic is quite complex, especially when it comes to making a stable membrane work in aggressive environments (very basic pH, presence of strong oxidants, acid/base cleaning cycles, etc.). It is important to know that the nomination of green membranes is very restrictive. Indeed, the 12 principles of green chemistry [187,188] must be applied to the preparation and use of membranes, as to any other system. The study of current manufacturing processes [189] shows that the polymers used industrially are not the most virtuous, either for their preparation or for the management of their end of life. Moreover, as with any separation technique, it generates residues that must be minimized and recycled in the best conditions. Initially, the solution envisaged was the recycling of solvents, but also the reuse of membrane materials at the end of their life [190,191,192]. Numerous research studies aim more radically at the replacement of solvents or the use of biosourced or degradable polymers such as Carrageenan, Cellulose, Chitosan, Cyclodextrins, Starch, Silk, Alginate, and Chitin [193,194,195]. The problem is very complex because it must reconcile a high resistance and a long life with the conditions of use required by the optimization of the operation and the cleaning procedures and because of the treated mixtures involving phenomena of ‘fouling’ or ‘scaling’ [41,162,163,196,197,198,199,200,201].

In the absence of a completely “green” synthesis method, it is possible to find surface or structural modification reactions of the membrane essential to its operation or to improve its performance in a notable way that respects the principles.

It should not be forgotten that if the membranes used do not meet the desired criteria, and using a membrane technique only makes an important contribution insofar as it makes it possible to replace much more polluting or energy-consuming processes.

So far, we have managed to find systems that can polymerize in water or sometimes use only a little ethanol. Chitosan and pectin are our two reference biosourced materials. However, the membranes obtained are not yet optimal and have not been published. There is still a lot of development to be done before reaching a pre-industrial membrane.

## 6. Conclusions

The research activity on membranes at the University Paris-Est Créteil (UPEC) dates back to the 1970s, when the university was founded. It has evolved a lot, and has been able to adapt to the evolution of the French and world socio-economic problems. This activity has covered many processes (pervaporation, diffusion dialysis, Donnan dialysis, neutralization dialysis, electrodialysis, electrolysis, nanofiltration) and several types of membranes (dense, nanoporous, ion-exchange, composites). The applications are also numerous and diverse (water and industrial effluent treatment, bleach production, selective lithium separation, green hydrogen production, etc.). The former collaboration with the State University of Kuban (Russia) and the recent collaborations with Qassim University and some Tunisian universities have been driving forces in the dynamics that characterize the membrane activity within the UPEC. The scientific record of this activity over the last decade is, as it was already for some decades, quite rich on all levels (doctoral supervision, scientific production, publications, industrial contracts, etc.). Our research perspectives are quite numerous and show a will to go further and to contribute to the development of membrane activity in France and in the world. We remain open to any national, European, or international collaboration that will allow us to go further in our international projects.

## Data Availability

The data presented in this study are available on request from the corresponding author.

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
