# Peer review of "Research on Membranes and Their Associated Processes at the Université Paris-Est Créteil: Progress Report, Perspectives, and National and International Collaborations"

_membranes, 2023, doi:10.3390/membranes13020252_

Round 1

Reviewer 1 Report

The manuscript is well written. I recommend it for publication after minor changes.

(i) Imrprove engligh throughout the manuscript.

(ii) Check spelling and typo error throughout.

Author Response

Dear Reviewer,

Please see attached our responses to your comments.

Best regards

Reviewer 2 Report

The manuscript submitted by Dammak et al reviewed the history and ongoing research of membranes for ion exchange, dialysis process, and recovery of water at the University Paris-Est Créteil. The manuscript is reasonably written with observable flaws. However, the bigger issues come with the literature it covered and limits in the review of critical findings on both technological and fundamental aspects.  The narrow focus of this review provides no broad impacts to the membrane community. The way how the collaborative research was established is not suitable for being part of the article on Membranes which focuses on advancing the understanding of membrane performance and structure. The extraction of Li+ is growing increasingly important in the research of renewable energy. The review provided in this manuscript did not include the efficient ligands and variance in the technical routes from any aspect. Also, the visions for the future research of membranes research has not been provided which is not commmon for a high-quality review article. 

Minors

1) If the figures are pasted from other publications, the permissions granted by the publishers need to be explicitly declared;

2) Page 3, line 101. The sentence is not complete;

Pge 3 127. Check the spelling for "Rhône Poulenc...";

Author Response

(The authors gave the same response as above.)

Reviewer 3 Report

The present manuscript introduces the history of membrane research in University Paris-Est Créteil (UPEC). This review clearly describes the membrane researches sorting with membrane applications and also shows the membrane projects and collaborative works. The referee undoubtedly thinks this review is meaningful for membrane society, and recommends to address the following minor points related to the manuscript format prior to publication.

1.     Create table of content or list of content before the introduction so that readers can easily find out the content of the revie at a glance.

2.     Make a Figure of chronical table or illustration that shows membrane researches and projects with a timeline. 

Author Response

(The authors gave the same response as above.)

Reviewer 4 Report

This is a nice review to show the research on membranes and associated processes at the University Paris-Est Créteil (UPEC). It is helpful for membrane researchers to know the UPEC, then some potential cooperation can be built up, which is beneficial for the development of membrane separation. Therefore, I suggest publishing this review paper after minor revision.

1. The resolution of figures needs to be improved, such as in Figure 1.

2. The absence of annotation and explanation of the three figures in Figure 12 can confuse the reader.

3. Format regularity also needs to improve! For example, Figure 11 has an outer border, but Figure 12 has no outer border.

Author Response

(The authors gave the same response as above.)

Reviewer 5 Report

A review paper written on "Research on Membranes and Their Associated Processes at The Université Paris-Est Créteil. Progress Report, Perspectives, and National and International Collaborations".

1. The paper does not adhere to the standard IMRAD structure. There is no rationale, no aim/goal, no method description, the results appear to be expanded throughout the paper, and there is no proper discussion. The paper is written in the style of a report.

2. The abstract needs to be rewritten. The abstract at this point appears to be the only part of the Introduction and objective. Despite the fact that it is a review paper, the abstract should summarise the entire paper (introduction, methods, results, conclusion/recommendation). The introduction sections should go into greater detail about the importance of membranes. In the introduction, the hypothesis of this article should be clearly stated. 

Author Response

(The authors gave the same response as above.)

Round 2

Reviewer 2 Report

The manuscript submitted by Baklouti and colleagues reviewed the research on membranes conducted at UPEC with ongoing research, perspective, and collaborations. The previous questions raised by reviewers have been answered with proper revisions to the manuscript and sufficient scientific support. The manuscript is well written with very few language issues. Therefore, the current form of the manuscript is qualified for publication on the journal Membranes.

Reviewer 5 Report

The questions raised by the reviewer have been addressed thoroughly therefore, the manuscript in its current form is appropriate for publishing in the journal "Membranes".